# Three-Dimensional Drone Exploration with Saliency Prediction in Real Unknown Environments

**Ming-Ru Xie** †, **Shing-Yun Jung** † and **Kuan-Wen Chen** *

Department of Computer Science, National Yang Ming Chiao Tung University, Hsinchu 30010, Taiwan
* Correspondence: kuanwen@cs.nycu.edu.tw
† These authors contributed equally to this work.

**Abstract:** In this paper, we propose a three-dimensional autonomous drone exploration system (ADES) with a lightweight and low-latency saliency prediction model to explore unknown environments. Several studies have applied saliency prediction in drone exploration. However, these studies are not sufficiently mature. For example, the computational complexity and the size of the developed prediction models have not been considered. In addition, some studies have only proposed saliency prediction models without actually applying them to drones. The ADES system proposed in this paper has a small and fast saliency prediction model and uses a novel drone exploration approach based on visual-inertial odometry to solve the practical problems encountered during drone exploration, such as collisions with and the repeated exploration of salient objects. The proposed ADES system performs comparably to the state-of-the-art, multiple-discontinuous-image saliency prediction network TA-MSNet and enables drones to explore unknown environments with high efficiency.

**Keywords:** automation; drone exploration; location awareness; prediction methods; UAV exploration

## 1. Introduction

Humans receive up to 80% of their information through vision every day; however, only a small amount of the information can be further processed by the visual system [1]. The advantage of this is that only useful information will be analyzed, and performing calculations on useless information can be avoided [1]. In the computer vision research, visual saliency refers to using algorithms that focus on the most visually distinct or important regions or objects in an image. This task is performed by imitating the human visual system. The traditional approach is to extract features from an image [2]. With the popularity of deep learning techniques, deep convolutional networks have also been applied to learn features from data [2]. Research in visual saliency detection can be grouped into two major directions: salient object detection and saliency prediction [1]. Salient object detection strive to find the most attention-grabbing objects in a scene [3]. Saliency prediction predicts which location in a scene attracts human attention the most [1].

Visual saliency detection has several applications, such as aerial robots [4], healthcare systems [5], facial expression recognition [6], and so on. One recently emerging application is drone exploration, i.e., drones autonomously identifying important objects in the environment for priority exploration based on visual saliency. Especially in complex and unknown environments, the entire environment can be explored quickly by detecting highly salient objects or areas [4] to avoid wasting time exploring unimportant ones.

The existing research on drone exploration using visual saliency has several limitations. Dang et al. [4] proposed a drone exploration application that applies a visual saliency prediction model; however, the application requires a considerable amount of time to construct a map to plan the exploration path. In addition, they used a saliency prediction model that does not consider the entirety of the environmental and temporal information.

Consequently, their system might not be efficient for exploration in unknown environments. The saliency prediction model proposed by Chu et al. [2] (MS-Net, multiple-image saliency network) was the first to consider the entirety of the environmental information. MS-Net uses multiple discontinuous images of the surrounding environment to predict saliency, but it does not consider the temporal information, which can potentially lead to inconsistent results in different time frames; thus, this model might have reduced efficiency in drone exploration systems. Huang et al. [7] proposed TA-MSNet (Temporally Aggregating Multiple-Discontinuous-Image Saliency Prediction Network), which is an improvement of MS-Net and accounts for the surrounding environment and temporal information. TA-MSNet uses the ConvLSTM module to retrieve time-series data between images over a certain time. However, TA-MSNet is relatively slow, and the size is relatively large, which makes it less suitable for application in practical drone exploration. Moreover, MS-Net and TA-MSNet have only been tested in simulations involving the use of image data sets; their feasibility for applications in drone exploration has not been tested in the field.

In this paper, we propose an autonomous drone exploration system (ADES) that integrates our proposed Lite-TA-MSNet [8] visual saliency model with a novel drone exploration approach based on visual-inertial odometry (VIO). Lite-TA-MSNet improves upon TA-MSNet, making it faster, lighter, and more suitable for real-time drone computing. The proposed drone exploration approach integrates saliency, inertial measurement unit (IMU) data, compass data, and visual simultaneous localization and mapping (SLAM). A drone that uses the proposed approach can quickly detect salient objects in unknown environments and record the location of the explored salient objects. In addition, the proposed approach does not require prior map construction, which increases its speed and exploration efficiency. Therefore, the proposed system enables a drone to (similar to human beings) observe an unfamiliar environment and then predict the saliency of objects on the basis of multiple discontinuous captured images for determining the exploration direction (Figure 1). Then, the drone explores salient objects, prevents drone–object collisions, and records explored locations until an entire environment has been explored.

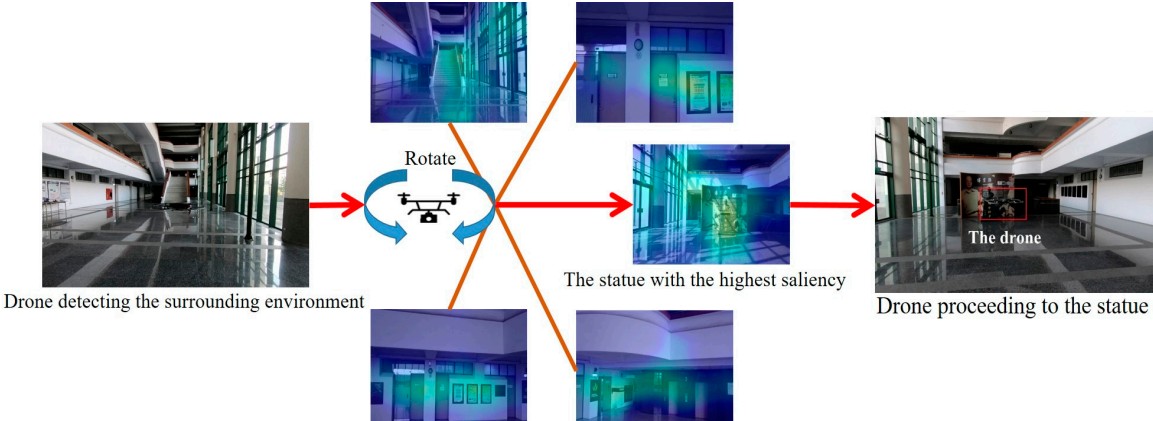

**Figure 1.** Overview of the proposed autonomous drone exploration system (ADES): the drone detects the surrounding environment and proceeds to a statue.

In summary, the key contributions of this paper are as follows:

1. We propose Lite-TA-MSNet, which is a faster, lighter, and more suitable model for real-time computation on drones than the present state-of-the-art TA-MSNet method.
2. We propose a novel drone exploration approach based on VIO to solve various practical problems encountered during exploration.
3. We propose a novel division strategy that involves dividing saliency maps into nine squares to enable drones to perform the three-dimensional (3D) detection of salient objects.
4. We integrate the proposed models and approaches into a 3D ADES.

## 2. Related Work

With improvements in drone technology, drones have increased in a range of applications [9]. For example, search and rescue operations [10] involve the use of drones to explore unknown and unfamiliar environments safely and quickly. Drone exploration in the real world is typically controlled by humans [2]. People who control a drone in an unknown environment get used to observing the whole environment, identifying objects or areas of interest, and then directing the drone to move toward a specific object or area of interest [2]. For many years, research has been conducted on the use of drones for the autonomous exploration of unknown environments [11,12]. Most of this research has involved maximizing the exploration of unknown environments [13] or mapping out exploration paths by using pre-established environment maps [13,14]. Therefore, the methods proposed in these studies are not very efficient for exploring unknown environments. To address these issues, research is being conducted on how to enable drones to mimic human exploration methods (visual saliency) for autonomously exploring unknown environments. Visual saliency models allow drones to skip visually unimportant areas, thereby reducing the exploration time and increasing exploration efficiency.

The first study to use a visual saliency model for autonomous drone exploration was that of Dang et al. [4]. They proposed a path planning strategy and path planner that explored unknown environments by detecting salient objects. In terms of experiments, in addition to simulating the method in the simulator, they also conducted flight exploration experiments in indoor environments. However, their system only allowed the drone to explore horizontally, and required considerable time to construct an environment map to plan the drone's exploration path. Chu et al. [2] proposed a model (MS-Net) that could predict the saliency after capturing multiple discontinuous images of the whole environment. Their system mimicked human exploration behavior. A human controlling a drone in an unknown environment first observes the environment before directing the drone to move toward an object or area of interest. The model of Chu et al. was the first to perform saliency prediction after considering the surrounding environment. The exploration behavior of MS-Net more closely resembles that of human behavior than do those of previous models. However, MS-Net only considers the environmental information. Furthermore, it has not undergone practical testing in drones; it has only undergone simulation tests with image data sets. Huang et al. [7] observed that MS-Net lacks temporal information, and improved upon MS-Net by having it incorporate environmental and temporal information, thereby improving the consistency of the prediction results in different time frames. However, the model of Huang et al. [7] has several practical limitations, such as a low speed and large size, and it has not been applied practically in drone exploration. Therefore, its performance in real-life applications remains unclear.

## 3. Architecture of the Proposed ADES

The architecture of the proposed ADES, which is a modification of the drone control system proposed in [15], is described in the following.

### 3.1. Hardware Architecture

As shown in Figure 2, a drone's images, IMU data, and compass data are transmitted through a USB cable to a remote controller and then transmitted from the remote controller to a smartphone (an ASUS ZenFone 6 in this study). The data are then transmitted through Wi-Fi from the smartphone to a laptop running the ADES. The laptop used in this study was an AMD Ryzen 7 7350H computer processing unit with 16 GB of RAM and an NVIDIA GeForce GTX 1660 Ti with Max-Q graphics card. The ADES controls the drone by sending it control commands through Wi-Fi after analyzing the transmitted data. A DJI Mavic Pro drone was used in our prototype system. This drone is lightweight and can be fitted with a propeller guard, which makes it relatively safe to fly indoors. A propeller guard prevents the drone from being damaged or causing damage to objects in the case of collisions.

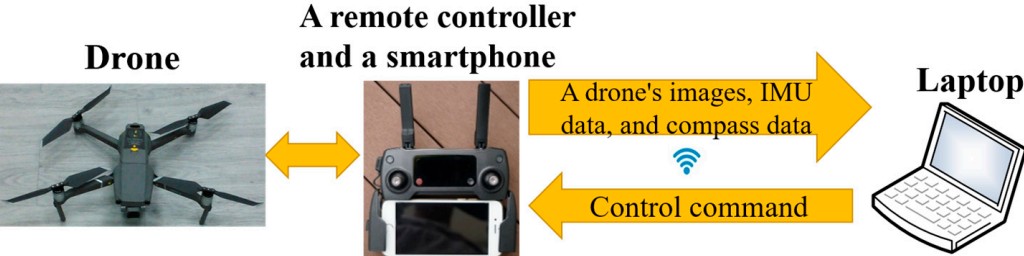

**Figure 2.** Hardware architecture of the proposed ADES.

### 3.2. Software Architecture

To transmit the drone's images, IMU data, and compass data from the smartphone to the laptop, we used a modified version of the Dronebridge application [15]. Dronebridge is an Android app based on the DJI Mobile software development kit (SDK) and the Android-VideoStreamDecodingSample application [16]. Raw images from the drone's camera are decoded using the Android-VideoStreamDecodingSample application, and the DJI Mobile SDK is used to get the IMU and compass data. A user datagram protocol (UDP) client program is embedded in the Dronebridge application that is responsible for transferring data to the ADES. A UDP server is embedded in the ADES to receive data.

The command generator of the ADES is responsible for converting the results in the data fusion into corresponding control commands. For example, in Section 4.1. Step 5, the drone needs to rotate to the specified drone heading angle. In this case, the command generator will generate the control command: pitch=0, roll=0, yaw=drone heading angle, and verticalThrottle=0. The data format of control commands: pitch (drone's pitch in m/s), roll (drone's roll in m/s), yaw (drone's yaw in degrees) and verticalThrottle (drone's ascending or descending speed in m/s) are from the DJI mobile SDK API.

The ADES is embedded in a transmission control protocol (TCP) client program to send the control commands to the Dronebridge application. A TCP server is embedded in the Dronebridge application, which allows this application to receive control commands from ADES. The control commands are a set of strings based on the DJI Mobile SDK that control the drone. These commands include move up, down, left, right, forward, and backward; turn left and right; take off; and land. Dronebridge parses the control commands, converts them into DJI Mobile SDK application programming interface parameters, and finally generates DJI control commands for flight control through the DJI Mobile SDK.

The architecture of the proposed ADES is shown in Figure 3. It contains two parts: the Lite-TA-MSNet model and a drone exploration module. Lite-TA-MSNet (Figure 4) is an improvement of TA-MSNet. Because the backbone of TA-MSNet, namely EML-Net, is relatively slow, we replaced it with EfficientNet-b0 [17] to accelerate the computational speed of the network. The reason for choosing EfficientNet is that its convolution operation is depth-wise separable convolution, which is designed to accelerate the CPU inference time of the model on the edge device. TA-MSNet uses a transformer for the saliency fusion of multiple images. The number of parameters in the transformer was considerably large resulting in TA-MSNet having up to 108.07 million parameters. Therefore, we replaced the transformer with a graph convolutional network [18]. The number of parameters then decreases to 57.48 million. Lite-TA-MSNet can easily analyze images captured by the drone in five directions and then predict and output saliency values for these images to the drone exploration module.

The drone exploration module is based on the VIO-based drone exploration approach in Section 4.1, which controls the drone moves in the direction with the highest saliency value among the five different directions based on the drone's images, IMU data, and compass data. The drone exploration module performs visual SLAM, conducts data fusion, and generates commands using a command generator. The visual SLAM software used by the proposed ADES is ORB-SLAM [19], which performs real-time SLAM according to the feature points in an image. In our prototype system, ORB-SLAM2 [20] is used because of its

stability. This algorithm can generate a sparse 3D scene from an input image and calculate the camera trajectory. In addition, it can be used by monocular cameras and in indoor and outdoor environments. The ADES inputs drone images in real time to ORB-SLAM2 to calculate and record the drone trajectory. The data fusion involved fusing the saliency values from Lite-TA-MSNet, IMU data, compass data, and the drone trajectory to the command generator. Finally, the flight control of the drone receives control commands from the command generator and used them to control the drone.

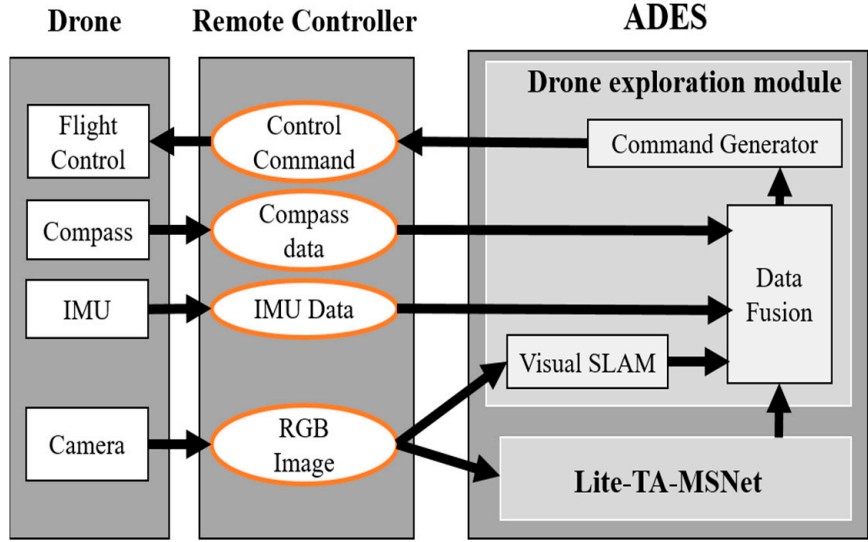

**Figure 3.** Architecture of the proposed ADES.

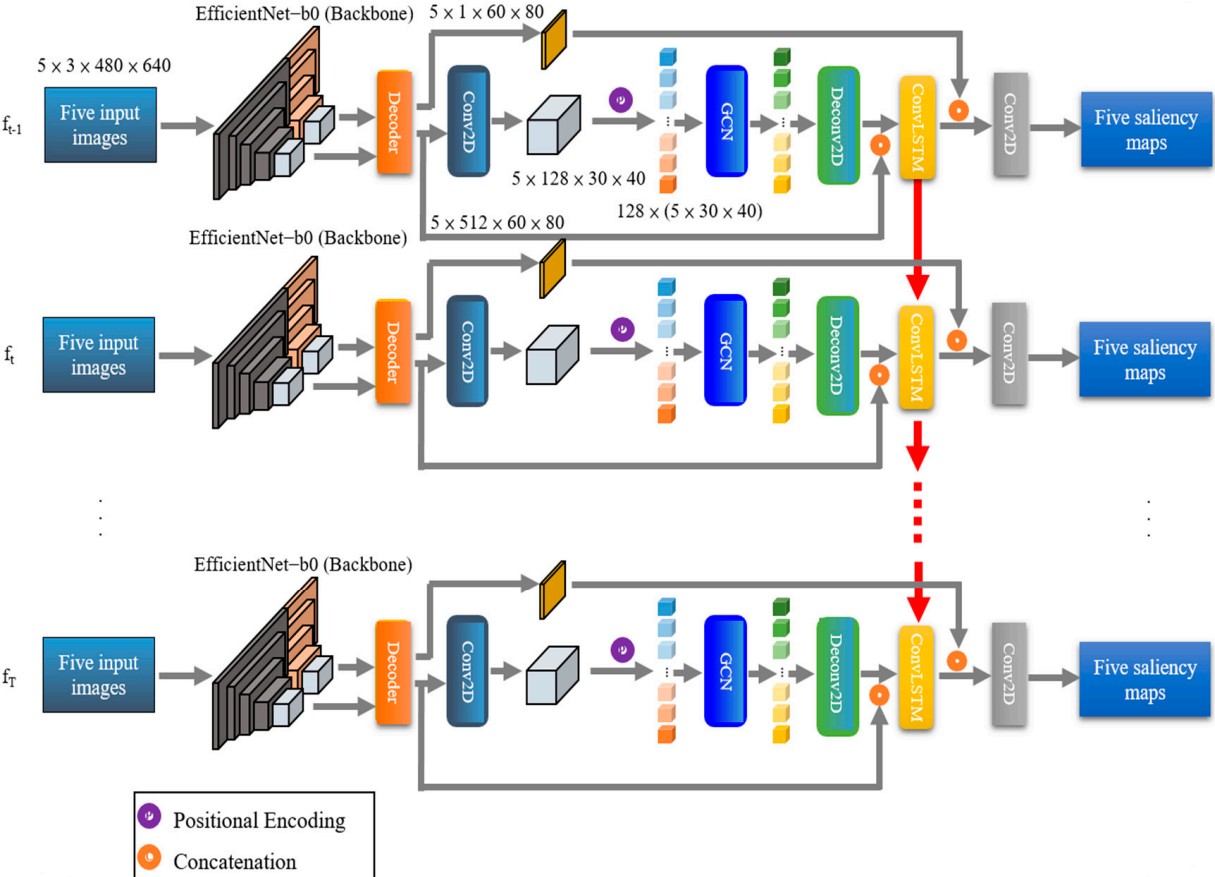

**Figure 4.** Architecture of Lite-TA-MSNet improved from TA-MSNet [7].

The drone exploration module is implemented in C and C++, and Lite-TA-MSNet is implemented in Python and PyTorch. According to the solution [21] on the official website of PyTorch, to load the PyTorch model in C++, this model must be converted into Torch Script. However, C++ and Python are different programming languages with different features. Therefore, using the official solution [21] might lead to some incompatibility issues between C++ and Python. For example, empty lists issue [22], scripting arguments call issue [23], and type annotation issue [24]. To avoid these problems, we used the TCP for data transmission between the drone exploration module and Lite-TA-MSNet instead of loading the PyTorch model in the drone exploration module. That is to say, Lite-TA-MSNet transmits the corresponding saliency values of the images captured by the drone in five directions to the drone exploration module through the TCP. In addition to solving the compatibility issues, others advantage of the aforementioned method are that Lite-TA-MSNet and the drone exploration module can be placed in separate servers and that data can be transferred over the network. That is, Lite-TA-MSNet can be placed in a server with a dedicated graphics processing unit, and the drone exploration module can be placed in another server.

The workflow of the ADES can be described as follows. The ADES first rotates the drone and observes the environment. Five images with different orientations and corresponding drone heading angles are acquired while the drone is rotating. These five images, live images from the drone, compass data (drone heading angles), and IMU data (drone's velocity) are then fed to the ADES. After receiving the analysis results from Lite-TA-MSNet, the drone exploration module directs the drone to turn and fly toward the object with the highest saliency value based on the object's saliency value and the corresponding drone heading angle. The images of the flight process are sent to ORB-SLAM2, and the drone exploration module compares the current flight trajectory output from ORB-SLAM2 with previously recorded trajectories. A high degree of similarity between current and previous trajectories indicates that the salient object has been explored. The system then stops the drone and directs it toward the object with the next highest saliency value. The drone stops when it reaches a salient object because it stops whenever it detects an obstacle. Commercial drones are typically equipped with obstacle avoidance systems. Our method of having the drone stop when it reaches a salient object involves the use of an obstacle avoidance system. When the ADES detects that the drone has almost stopped (the drone's velocity approaching 0), it rotates the drone and captures five images of the surrounding environment to begin a new round of salient object detection. This cycle continues until the drone has explored the entire environment.

## 4. Proposed Methods of ADES for Drone Exploration

We propose the following methods to improve the accuracy of drone exploration.

### 4.1. VIO-Based Drone Exploration Approach

The VIO-based drone exploration approach runs on the drone exploration module to receive and transmit data from drones and control drones to explore unknown environments. The main steps of this algorithm are as follows:

1. The drone is directed to rotate by 72° five times and capture one image and its corresponding drone heading angle after each rotation.
2. After five rotations, the five captured images are transferred to Lite-TA-MSNet. Compass data (drone heading angles) of the five images are transferred to drone exploration module.
3. Before Lite-TA-MSNet analyzes the five images, data fusion uses a map data structure to fuse the file names of the five images and their corresponding drone heading angles. A map is a data structure that stores the key and the corresponding value of the key. Data fusion stores the file names of the five images and the corresponding drone heading angles of each image into the map. After Lite-TA-MSNet finds the image with the highest saliency value, the file name of the image with the highest saliency

value is output. Next, data fusion finds the corresponding drone heading angle from the map according to the file name of the image with the highest saliency value.

4.  The drone heading angle corresponding to the image with the highest saliency value is determined.
5.  The drone is rotated to the drone heading angle in Step 4.
6.  The drone is directed to move forward at the drone heading angle in Step 4. RGB images are sent from the drone to the visual SLAM system to record the explored drone trajectory.
7.  When the drone's velocity is detected to be close to 0, the drone is stopped, and Step 1 is returned to.

### 4.2. Max Saliency Clustering

The highest saliency value in the current saliency map is calculated by only finding a single pixel with the highest saliency value in an image. However, this method is prone to interference from light and reflections, which results in object saliency being determined by a single pixel with the highest brightness, rather than the object itself. To mitigate brightness interference, our method involves using a clustering algorithm. Because the number of clusters to be set is not known in advance and noise must be handled automatically, the DBSCAN algorithm [25] is used to find clusters with similar saliency values. The cluster with the highest average saliency is considered to have the maximum saliency in a particular saliency map. In this way, the highest saliency in an image is not determined by a single pixel.

### 4.3. Nine-Square Division Strategy for Saliency Maps

Drones are lightweight, highly maneuverable, and capable of taking off and landing vertically and hovering in the air similar to a traditional helicopter; therefore, drones are suitable for exploration in unknown environments. Unfortunately, current drone exploration studies [4] can only explore unknown environments in a two-dimensional (2D) manner (in a receding horizon manner). However, drones can move in three dimensions. A drone's mobility is not fully utilized in 2D exploration. Therefore, we propose a nine-square division strategy, which involves dividing the saliency maps obtained from Lite-TA-MSNet into nine parts, to address this problem (Figure 5). The part of the saliency map with the highest saliency value is determined, and the drone is subsequently directed to move to this area. For example, in Figure 5, the left side of the image has the highest saliency value; thus, the drone is directed to move to the left. The drone is continuously adjusted with the left side as the center to ensure that it does not deviate from the left side while moving.

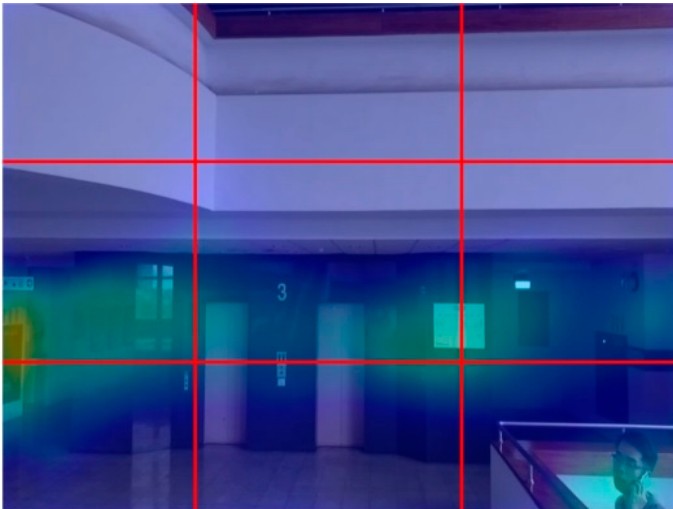

**Figure 5.** Application of the proposed nine-square division strategy on an image.

## 5. Experiment

The TA-MSNet dataset was derived from the 17 "camera moving" videos in the VR-EyeTracking dataset [26]. The reason for choosing "camera moving" videos is that the camera moves while shooting the video, similar to as if a drone is exploring the scene. After these videos are selected, the unnecessary noise is further removed. Finally, five sub-images are randomly cropped from each frame in each video to build the TA-MSNet dataset. Thirteen videos of the TA-MSNet dataset are used for training and the remaining videos are used for testing. TA-MSNet and the proposed model were trained, tested, and evaluated using the same data set and metrics. Regarding the CPU inference time test, TA-MSNet and the proposed model were performed on the same machine (Intel(R) Core(TM) i9-10900X CPU @ 3.70GH).

The metrics used to evaluate the performance were similarity (SIM), correlation coefficient (CC), and Kullback–Leibler divergence (KL). SIM [27] is a metric used to measure the similarity between two saliency maps. The maximum value of SIM metric is 1 and the minimum value is 0. The CC [28] is a statistical indicator that has been be used to assess the correlation between two variables. The CC ranges from -1 to 1. The larger the positive number, the more positive the correlation, and the smaller the negative number, the more negative the correlation. A value of 0 means no correlation between the two variables. Assuming that there are two independent probability distributions P, Q in the same sample space, the Kullback-Leibler divergence [29] is an indicator to assess the degree of difference between P and Q.

We compared the proposed model with the state-of-the-art method (TA-MSNet) [7]. Table 1 presents the results of the comparison in three aspects. The proposed model was 6% faster than was TA-MSNet about the CPU inference time, and had 47% fewer parameters than did TA-MSNet. The comparison of the performance metrics shows that the proposed model and TA-MSNet had similar performance. Additional experimental details can be found in [7].

**Table 1.** Comparison between proposed Lite-TA-MSNet and TA-MSNet.

|  | **Proposed** | **TA-MSNet** |
| --- | --- | --- |
| CPU Inference Time (s) | 6.0 | 6.4 |
| Number of Parameters (million) | 57.48 | 108.07 |
| CC ↑ | 0.2 | 0.21 |
| SIM ↑ | 0.25 | 0.25 |
| KL ↓ | 1.22 | 1.23 |

To demonstrate the feasibility of ADES in the real world, we tested the ADES inside a building on the NYCU campus. The experimental environment was spacious and contained various objects, including a statue, a stairway, doors, and fire extinguishers. We considered this environment ideal for indoor testing purposes. A demo video of the application of the ADES can be found at the following URL for ref. [30].

The demo video shows an example of the drone exploring the room. After taking off, the drone rotated to observe its surroundings and captured five photos (Figure 1). The five images, IMU data, and compass data were transmitted to the ADES, which analyzed the five images and identified the object with the highest saliency value and its corresponding drone heading angle. The ADES then directed the drone to move toward the statue, which was the object with the highest saliency value. The obstacle avoidance system detected the obstacle (the statue) and stopped the drone before it reached the statue, which then implied that the statue had already been explored. Next, the drone rotated in front of the statue and then started a new round of exploration. In this new round of exploration, the object with the highest saliency value was the stairway, and the drone proceeded toward the stairway.

## 6. Conclusions

In this paper, we propose a lightweight and low-latency saliency prediction model and a novel drone exploration approach based on VIO for the implementation of a 3D autonomous drone exploration system that does not require preconstructed environment maps. The purpose of this system is to enable a drone to perform automatic exploration while considering the saliency of objects, considering the environmental and temporal information, and solving the problems typically encountered during exploration. The proposed model, namely Lite-TA-MSNet, is faster, lighter, and more suitable for drone exploration than the state-of-the-art multiple-discontinuous-image saliency prediction network TA-MSNet method. We also propose a VIO-based drone exploration approach and a nine-square division strategy to improve the performance of the proposed system. The proposed system provides a practical framework for drones to conduct autonomous exploration in unknown environments. Although the system was only tested indoors, the experimental results demonstrated its feasibility. The ADES can explore complex unknown environments and prevent people from encountering hazards. For example, exploring in an abandoned factory. Future research will focus on making the proposed model lighter, faster and more efficient, and will test the model in a wide variety of unknown environments.

**Author Contributions:** Conceptualization, M.-R.X., S.-Y.J. and K.-W.C.; methodology, M.-R.X., S.-Y.J. and K.-W.C.; software, S.-Y.J. and M.-R.X.; validation, S.-Y.J.; formal analysis, S.-Y.J.; data curation, S.-Y.J.; writing—original draft preparation, M.-R.X.; writing—review and editing, M.-R.X. and K.-W.C.; supervision, K.-W.C.; project administration, M.-R.X. All authors have read and agreed to the published version of the manuscript.

**Funding:** This research was funded by National Science and Technology Council of Taiwan grant number 111-2634-F-A49-010- and 111-2628-E-A49-003-MY2.

**Data Availability Statement:** Not applicable.

**Conflicts of Interest:** The authors declare no conflict of interest.

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
