# Peer review of "Three-Dimensional Drone Exploration with Saliency Prediction in Real Unknown Environments"

_aerospace, doi:10.3390/aerospace10050488_

Round 1

Reviewer 1 Report

1 - The sentences in the paper should be revised carefully for a better and more coherent paper. For example

Line 13 instead of for example use e.g.

Line 118, “… is described in the following text” should be “…. is described in the following”

2- For section 3.1 hardware agriculture, a diagram that shows all the connection can be more helpful, similar to https://github.com/NVIDIA-AI-IOT/redtail/wiki/Skypad-TBS-Discovery-Setup (7th picture)

3- In line 163, the drone exploration module, it is expected to explain more about how the data fusion is handled and what is the algorithm of command generator.

4- Eliminate lines 117-182 from “However, to …type annotation” and briefly explain that due to incompatibility between C++ and Python some modifications are needed.

5- In line 273, the link for the video should be either in a footnote or referred as a reference.

6- At least some results comparing your method with TA-MSNet in terms of dynamical behavior can be presented. Also, the drone mission and some graphs with results can improve the paper.

Reviewer 2 Report

The authors propose a UAV-based exploration methodology by applying a saliency prediction model (improved TA-MSNet), showing better performance with respect to other strategies. The topic is of great interest for the UAV/UAS community, and the paper is appropriate for the needs of a journal like Aerospace (or Drones, perhaps). Except for some typos, English style and grammar are ok, the review of the literature is adequate, and the most relevant pieces of previous work are correctly cited. Nevertheless, the paper needs, in my opinion, some further thinking in terms of structure and description of the method, results analysis and applicability of the results. In the following I am listing some considerations that could help improving the scientific quality of the paper.

1.       In order to focus the reader’s attention to the problem of visual saliency prediction, I would suggest an additional reference focusing on (and reviewing) the most important achievements and methodologies (Yan, F.; Chen, C.; Xiao, P.; Qi, S.; Wang, Z.; Xiao, R. Review of Visual Saliency Prediction: Development Process from Neurobiological Basis to Deep Models. Appl. Sci. 2022, 12, 309. DOI: 10.3390/app12010309). Moreover, I would suggest explaining the acronyms MS-Net and TA-MSNet (Temporally-Aggregating Multiple-Discontinuous-Image Saliency Prediction Network).

2.       Some details on the saliency prediction model used (or some references pointing to the model used) will be useful for a better understanding of the overall methodology. In particular, the differences between saliency computation and object detection should be pointed out, and the type of detected saliency should be specified (does the algorithm rely on image features and statistics to localize the most “interesting” sections of an image? Does the algorithm rely on frame-by-frame analysis and mark as “salient” object that move? Does the algorithm perform some sort of “bounding” of regions of high saliency? What if two images in the same exploration stage (i.e., after having collected five images in a station of the drone) have the same saliency? How is the saliency quantified (on a pixel-by-pixel basis, as a matrix of numbers representing the saliency of a region of the frame, as a “visual” saliency map, etc.)? Could some “artificial” features (for example, shadows or spotlights) impair the saliency prediction model?

3.       A definition of the metrics used to evaluate the algorithm performance (SIM, CC, KL) could add clarity to the described methodology. In Table 1, results with four or five significant digits are meaningless. I would propose to use two or three figures (e.g. 1.22 instead of 1.2218, 0.25 instead of 0.2483 and so on). What do the arrows near CC, SIM and KL mean? What do the author mean by “number of parameters”? Is it the number of analyzed pixels? What is the resolution of the images (the clustering of similar saliency values could group different salient objects in one single cluster, depending on the image resolution and the lighting conditions)? Is it sufficient to produce “reliable” saliency maps? Can a 6% improvement of CPU time with respect to TA-MSNet justify the name “Faster TA-MSNet” of the authors’ algorithm (I would expect a computational burden reduced by 20% at least)?

4.       The demo video shows an example of 2D exploration, since the drone is moving at constant distance from the floor. The 3D strategy proposed in Sec. 4.3 does not seem to be implemented in experimental trials.

5.       Point 2 of he VIO approach is not clear. If the UAV acquires an image, then rotates by 72 degrees, acquires another image, and so on up to 360-degree rotation, and is practically standing still during the acquisition of the five images, why the authors need to acquire “IMU data (drone’s velocity) of the five images” (line 221)?

6.       Line 262. What data set were used to evaluate the performance of TA-MSNet and “faster” TA-MSNet?

Reviewer 3 Report

In this article, the authors address current and interesting issues related to UAV exploration. The presented results are interesting and indicate the potential of the work undertaken to modify the saliency prediction network TA-MSNet method. 

However, the article does not detail the improvements to the algorithm, which is expected in a scientific article. Otherwise, it resembles press material of a commercial offer.

In conclusion, the area of the article should be expanded with a detailed description of the improvements to the saliency prediction network TA-MSNet method. 

The comparison of the developed method with the TA-MSNet method is, in my opinion, also insufficiently supported by results and information on test conditions, test assumptions, comparison factors, measurement technique, etc.

Round 2

Reviewer 2 Report

The authors have correctly addressed my concerns, modifying accordingly the text. The scientific soundness of the research is improved, and in my opinion the paper is now ready for publlication.

Reviewer 3 Report

The additional implementation details added to the article positively influenced the quality of the research presented.